# Introgression of Resistance to Leafminer (*Liriomyza cicerina* Rondani) from *Cicer reticulatum* Ladiz. to *C. arietinum* L. and Relationships between Potential Biochemical Selection Criteria

**Nesrine Chrigui** [1], **Duygu Sari** [1], **Hatice Sari** [1], **Tuba Eker** [1], **Mehmet Fatih Cengiz** [2], **Cengiz Ikten** [3] **and Cengiz Toker** [1,*]

1 Department of Field Crops, Faculty of Agriculture, Akdeniz University, Antalya TR 07070, Turkey; chrigui_nes@yahoo.com (N.C.); duygusari@akdeniz.edu.tr (D.S.); haticesari@akdeniz.edu.tr (H.S.); ekertuba07@gmail.com (T.E.)
2 Department of Agricultural Biotechnology, Faculty of Agriculture, Akdeniz University, Antalya TR 07070, Turkey; fcengiz@akdeniz.edu.tr
3 Department of Plant Protection, Faculty of Agriculture, Akdeniz University, Antalya TR 07070, Turkey; cikten@akdeniz.edu.tr
* Correspondence: toker@akdeniz.edu.tr; Tel.: +90-242-310-24-21 or +90-537-543-10-37

**Abstract:** The chickpea leafminer, *Liriomyza cicerina* (Rondani), is one of the most destructive insect pests of cultivated chickpea (*Cicer arietinum* L.) in the Mediterranean region under field conditions. For sustainable and environmentally friendly chickpea production, efforts have been devoted to managing the leafminer via decreasing the use of insecticides. Breeding of new resistant varieties is not only an efficient and practical approach, but also cost-effective and environmentally sensitive. To improve resistant varieties, breeders need reliable biochemical selection criteria that can be used in breeding programs. The first objective was to investigate the possible introgression of resistance to the leafminer from *C. reticulatum* Ladiz. (resistant) to *C. arietinum* (susceptible), then, to estimate inheritance of resistance to the leafminer for efficient breeding strategies, and finally, to study organic acid contents as selection criteria. Recombinant inbred lines (RILs) and their parents were evaluated using a visual scale of 1–9 (1 = free from leafminer damage and 9 = mines in more than 91% of the leaflets and defoliation greater than 31%) in the field under natural infestation conditions after the susceptible parent and check had scores of >7 on the visual scale. Superior RILs were found for resistance to the leafminer, and agro-morphological traits indicating that introgression of resistance to leaf miner from *C. reticulatum* to *C. arietinum* could be possible using interspecific crosses. The inheritance pattern of resistance to the leafminer in RILs was shown to be quantitative. Organic acids, including oxalic, malic, quinic, tartaric, citric and succinic acids in RILs grown in the field under insect epidemic conditions and in the greenhouse under non-infested conditions were detected by using high performance liquid chromatography (HPLC). In general, organic acids were found to be higher in resistant RILs than susceptible RILs. Path and correlation coefficients showed that succinic acid exhibited the highest direct effects on resistance to the leafminer. Multivariate analyses, including path, correlation and factor analyses suggested that a high level of succinic acid could be used as a potential biochemical selection criterion for resistance to leafminer in chickpea. Resistant RILs with a high seed yield resembling kabuli chickpea can be grown directly in the target environments under leaf miner infestation conditions.

**Keywords:** chickpea; *Cicer arietinum*; *Cicer reticulatum*; leafminer; *Liriomyza cicerina*; organic acids; succinic acid; path analysis

## 1. Introduction

Among insect pests, leafminers in the genus *Liriomyza* Mik. (Diptera: Agromyzidae) are well-known phytophagous insects that cause considerable economic losses worldwide

in the most important plant species used for the human diet [1,2]. In the Mediterranean region, the chickpea leafminer (*Liriomyza cicerina* Rondani) is one of the most devastating insect pests on chickpea (*Cicer arietinum* L.) production under field conditions [2,3]. Adult flies of chickpea leafminer feed on plant sap, while larvae of the leafminer feed on the mesophyll tissues of leaflets [2]. The ovipositors, females of the leafminer, perforate the leaflet epidermis to lay 1–30 eggs. After a 4 day incubation period, the larvae create whitish mines in the leaflets along the parenchyma tissue [2–4]. Under intensive infection conditions, these mines created by the leafminer reduce the photosynthetic area, which causes leaflets to fall. In severe leaf miner infestation, yield reduction in chickpea was reported to be as high as 40% [2].

The control of chickpea leaf miner is most effective when using insecticide weapons to deal with these problems, but their substances have harmful consequences on the environment such as the accumulation of residues and soil/water pollution. The widespread use of insecticides, the appearance of resistance mechanisms in insects, and the fact that many of these synthetic compounds have a broad spectrum of action can result in damage to non-target organisms. In view of these drawbacks, it is important to find alternative solutions such as cultural with biological controls and host plant resistance [5,6].

Several research studies on resistance to the leafminer have been carried out from the 1970s to date [6–9]. A total of 9500 germplasm resources of cultivated chickpea were screened for resistance to leaf miner and only a few resistant resources were identified [2,8]. Since there were not enough genetic resources for resistance to leaf miner in cultivated chickpea, about 200 annual wild *Cicer* species were screened and considerably resistant accessions of *C. bijugum* K.H. Rech., *C. echinospermum* P.H. Davis, *C. pinnatifidum* Jaub. Et Spach., *C. judaicum* Boiss., *C. chorassanicum* (Bunge) M. Pop. and *C. reticulatum* Ladiz. were found [9,10]. Some mutant *Cicer* lines were reported to be resistant to leaf miner [6,10–12]. Resistant cultivars can be bred by inter- and intra-specific crosses and mutation breeding. To breed for resistance to chickpea leaf miner, inheritance should be elucidated as the first step.

The chickpea secretes organic acids from its green parts including leaflets, leaves, stems, sepals and pods as a mechanism of defense against insect pests. There is an important genotypic variation in the proportion of secreted acids and the exudate contains mainly malic and oxalic acids [6,13,14]. A relationship between resistance to insect pests or diseases and organic acids has been revealed [2,14–17]. However, collection of organic acids in glandular hairs of chickpeas is time consuming and it is not a feasible or easily accessible method. In contrast, the useful method and easy approach, while less time-consuming, is to use organic acids from the leaf, leaflet, stipule and stem of plants [14]. Organic acids and secondary metabolites are considered to play a crucial role as biological protectants in chickpea [2,18–20]. Revealing relationships between organic acids and resistance to the chickpea leafminer will be beneficial for breeders in order to select resistant genotypes and improve resistant cultivars.

This paper aimed for introgression of resistance to the leafminer from *C. reticulatum* (resistant) to *C. arietinum* (susceptible), estimation of broad sense heritability for resistance to leafminer for well-organized breeding programs, inspection of potential selection criteria among oxalic, malic, quinic, tartaric, citric and succinic acids for resistance to the leafminer, and determination of direct and indirect relationships between organic acids and resistance to leafminer.

## 2. Materials and Methods

### 2.1. Plant Materials

A total of 130 recombinant inbred lines (RILs) derived from interspecific crosses between cultivated chickpea (*C. arietinum*, CA 2969, susceptible to the leafminer) and its progenitor (*C. reticulatum*, Akdeniz University wild *Cicer* (AWC) 602, resistant to the leafminer) were screened for resistance to leaf miner under field conditions. Both parents were homozygous, and three $F_2$ populations were produced by these parents. One of these

$F_2$ populations, having 119 $F_2$ plants, was advanced using single seed descent, that is, a seed was harvested each $F_2$ plant. RILs were advanced one generation per year up to $F_6$. A susceptible commercial variety (Sierra) grown in the USA, with a unifoliolate (simple) leaf, was used as a check in field experiments. Some studies revealed that larger leaflets in chickpea were preferred by leaf miner [6,10,11]. Figure 1 shows leaves of the parents and check Sierra. Eight resistant and eight susceptible lines selected from 130 RILs were grown both in the field under insect infestation and the greenhouse under non-insect infested conditions. These resistant and susceptible lines were used for organic acid analyses.

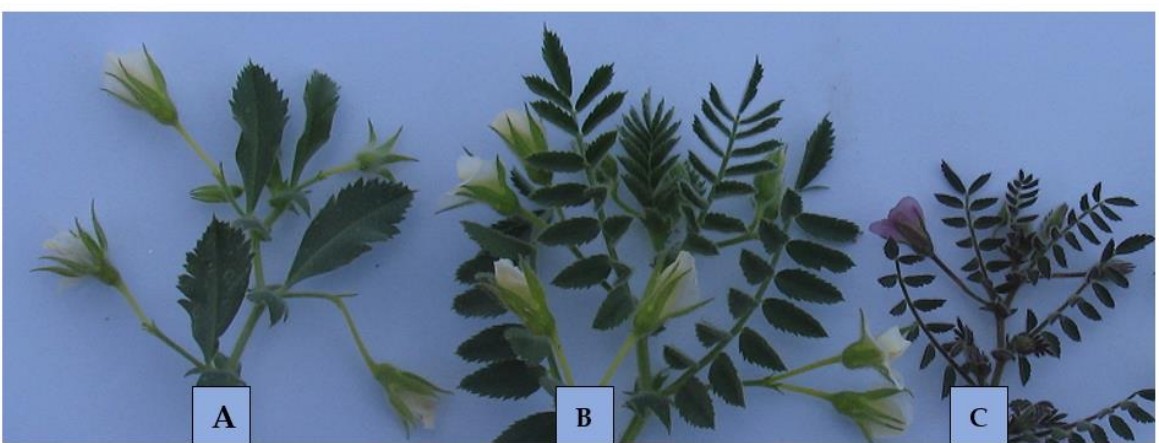

**Figure 1.** The leafminer-susceptible check Sierra (**A**), parents used in interspecific crosses between *C. arietinum* (CA 2969) (**B**) and *C. reticulatum* (AWC 602) (**C**).

### 2.2. Description of the Location

Field experiments were carried out in the experimental fields (30° 44′ E, 36° 52′ N, 51 m asl) of Akdeniz University, Antalya, Turkey for two years in 2017 and 2018. The growing season was arranged according to heavy infestations of the leafminer. Plants were, therefore, grown between February and July since damage by the leafminer can be seen in March. After field experiments, eight resistant and eight susceptible RILs to the leafminer were grown for organic acid analyses in the greenhouse under non-infested conditions and the field under insect infested conditions at the same location.

Soil properties of the experimental area and greenhouse were given by Ceylan et al. [21]. Organic matter, nitrogen, iron and zinc were found to be low due to the high pH of 7.96 and $CaCO_3$ of 26.5% [21,22]. Rainfall distribution was irregular during the vegetative stage and insufficient during the pod setting stage. Total rainfall in 2017 and 2018 was recorded as 605.9 mm and 954 mm, respectively. Plants in the open field were subjected to temperatures as high as 30 and 40 °C during flowering and pod filling stages, respectively, while an attempt was made to prevent drought with irrigation in the greenhouse [22,23].

### 2.3. Experiments, Agronomic Practices and Monitoring of Leafminer

In the field, experiments were set up in a randomized complete block design (RCBD) with two replications, whereas the experiment in the greenhouse was carried out in a RCBD with three replicates. In each replication, 40 plants of each RIL were planted in a single row of 2 m length with intra-row spacing of 50 cm and plant spacing of 5 cm. A planting of the susceptible check Sierra was repeated after 10 RILs.

Plants were neither fertilized nor irrigated in the field under insect infested conditions, while they were watered with drip irrigation in the greenhouse under non-infested conditions. Weed control was done manually at the vegetative stage.

RILs, their parents and the susceptible check Sierra were sown by hand in February to make the epidemic insect population coincide with the period prior to flowering time of chickpea. Thus, the leafminer infestation was guaranteed. Monitoring of the leafminer depended on the susceptible check Sierra and susceptible parent CA 2969.

### 2.4. Screening for Resistance to the Leafminer

RILs were screened for resistance to the leafminer (RL) in the field under natural infestation conditions according to a visual 1–9 scale (Table 1) originally suggested by Singh and Weigand [8]. In this scale, 1 is free from leafminer damage, and 9 is very highly susceptible with many mines in more than 91% of the leaflets and defoliation of more than 31%. Screening scores between 1 and 4 were resistant, whereas they were categorized as susceptible between 5 and 9.

**Table 1.** Visual 1–9 scale for resistance to chickpea leafminer in RILs derived from interspecific crosses between *C. arietinum* and *C. reticulatum*.

| Score | Response to Leaf Miner | Appearance of Plants |
|---|---|---|
| 1 | Very Highly Resistant | Free from any damage |
| 2 | Highly Resistant | Mines in less than 10% of the leaflets after careful observation |
| 3 | Resistant | Mines in less than 11–20% of the leaflets, no defoliation |
| 4 | Moderately Resistant | Mines present in 21 to 30% of the leaflets, no defoliation |
| 5 | Less susceptible | Mines present in 31 to 40% of the leaflets, some defoliation |
| 6 | Moderately susceptible | Mines in 41 to 50% of the leaflets, defoliation of 10% |
| 7 | Susceptible | Mines in 51 to 70% of the leaflets, defoliation 11–20% |
| 8 | Highly susceptible | Mines in 71 to 90% of the leaflets, defoliation 21–30% |
| 9 | Very highly susceptible | Mines in more than 91% of the leaflets and defoliation greater than 31% |

### 2.5. Agro-Morphological Traits

Growth habit (GH), as erect and spreading, and number of pods per node (PA), as single and double pods, were recorded. Leaves per plant (LP), leaflets per leaf (LL), leaflet width (LW), leaflet length (LH), plant height (PH), number of pods per plant (PP), biological yield (BY), seed yield (SY) and 100-seed weight (SW) were recorded as quantitative traits for the evaluation of relationships between resistance to the leafminer and agro-morphological traits.

### 2.6. Estimation of Heritability

The broad-sense heritability ($h^2$) for leaf miner resistance was estimated according to the ratio of genotypic variance ($\sigma_g^2$) to phenotypic variance $\sigma_p^2$ [24]:

$$h^2 = (\sigma_g^2/\sigma_p^2) \times 100, \; \sigma_p^2 = \sigma_g^2 + (\sigma_{gy}^2/y) + (\sigma_e^2/ry),$$

where $\sigma_{gy}^2$ and $\sigma_e^2$ are interaction variance for genotype by year and error, respectively, while $y$ and $r$ are number of year and replication, respectively.

According to distribution for resistance to the leafminer, the inheritance pattern was estimated in RILs. Similarly, the recessive or dominant nature of genes conferring RL was predicted on the basis of majority of susceptible and resistant RILs.

### 2.7. Analyses of Organic Acids by High-Performance Liquid Chromatography (HPLC)

Leaf samples were collected at night during the vegetative stage to minimize environmental stresses. Organic acids were analyzed in leaf samples. A quantity of 50 g of leaves was cleaned with deionized water, then dried at 40 °C. The leaf samples were ground with an iron mortar then 1 g was weighed. After that, the samples were blended with 10 mL of 0.2% $KH_2PO_4$ (pH level of the diluted solution was adjusted by $H_3PO_4$ at 3.9). The tube was transferred to an ultrasonic bath at room temperature for 15 min. At the end of the

ultrasonic treatment, the solution was centrifuged at 5000 rpm for 10 min and filtered through a Whatman No. 1 paper filter. Subsequently, the supernatant was filtered through a 0.45 μm polyvinylidene fluoride (PVDF) membrane filter. A volume of 0.5 mL of clarified solution was diluted with 1 mL of methanol in an HPLC vial. Ten μL of prepared dilution was then injected into a loop injection valve of high performance liquid chromatography (HPLC) (Agilent 1200 Technologies, Böblingen, Germany) in which a quaternary pump and diode array detector (DAD) were connected to the system controller. The HPLC column was C18-filled Agilent brand (4,6 mm × 250 mm, 5 μm) and the mobile phase was 0.2% $KH_2PO_4$ (pH level of the mobile phase was 3.9) containing 1 l of water and 1 mL of phosphoric acid with 1 mL/min flow rate at ambient temperature. The standard and samples in HPLC were performed at 210 and 214 nm. In order to quantitatively determine various peaks, a comparison of the integration area values of different standards with known concentration was performed with the sample peaks and the organic acid contents were calculated accordingly. Organic acid analyses were used prior to the present study [14,15].

*2.8. Statistical Analyses*

Descriptive statistics, correlations and analysis of variance (ANOVA) were performed for agro-morphological traits and organic acids. Means were compared via a Tukey's multiple comparison test. RL data were converted to percentages of mines. Path coefficients by Dewey and Lu [25] were studied to show direct and indirect relationships both between agro-morphological traits and resistance to the leafminer (RL) and between organic acids and resistance to the leafminer. Factor analysis by Cattel [26] was carried out with organic acids and leafminer resistance data sets.

## 3. Results

*3.1. Resistance to the Leafminer*

The effect of genotypic differences in resistance to the leafminer (RL) was statistically significant ($p < 0.01$), but genotype by year interaction was not for the same trait ($p < 0.05$). Scores for RL ranged from 3 to 9 on the scale (Table 2). A total of 130 RILs, their two parents and the check Sierra were identified to be neither "very highly resistant" nor "highly resistant" to leaf miner. However, three RILs (RILs 39, 56 and 72) were determined to be "resistant" to leafminer with a score of three on the visual scale (Figure 2). Twelve RILs were "moderately resistant" with scores of four on the scale. The male parent *C. reticulatum* was found to be "resistant" to leafminer, having a 3.5 score, while the female parent CA 2969 was "susceptible" to the pest, having a score of 7.5 on the scale (Figure 2).

**Table 2.** Means, standard errors (SE) and ranges for resistance to the leafminer and agro-morphological traits, and correlations between RL and agro-morphological traits in RILs derived from interspecific crosses between *C. arietinum* and *C. reticulatum*.

| Traits | Mean ± SE | Minimum | Maximum | Correlations |
|---|---|---|---|---|
| Resistance to leaf miner (RL) | 6.15 ± 0.8 | 3.00 | 9.00 | |
| Leaves per plant (LP) | 89.15 ± 1.5 | 31.00 | 176.50 | −0.141 * |
| Leaflets per leaf (LL) | 12.25 ± 0.6 | 10.00 | 14.00 | 0.018 |
| Leaflet length (LH) | 0.86 ± 0.01 | 0.50 | 1.30 | −0.267 ** |
| Leaflet width (LW) | 0.49 ± 0.01 | 0.30 | 1.00 | −0.234 ** |
| Plant height (PH) | 31.82 ± 0.3 | 21.50 | 48.00 | −0.314 ** |
| Pods per plant (PP) | 23.37 ± 0.7 | 6.00 | 63.00 | −0.071 |
| Biological yield (BY) | 7.27 ± 0.3 | 1.40 | 25.68 | −0.076 |
| Seed yield (SY) | 3.46 ± 0.1 | 0.25 | 11.81 | −0.213 ** |
| 100-seed weight (SW) | 16.76 ± 0.3 | 4.83 | 29.24 | −0.092 |

* and ** mean that correlation coefficients are significant at $p < 0.05$ and $p < 0.01$ levels, respectively.

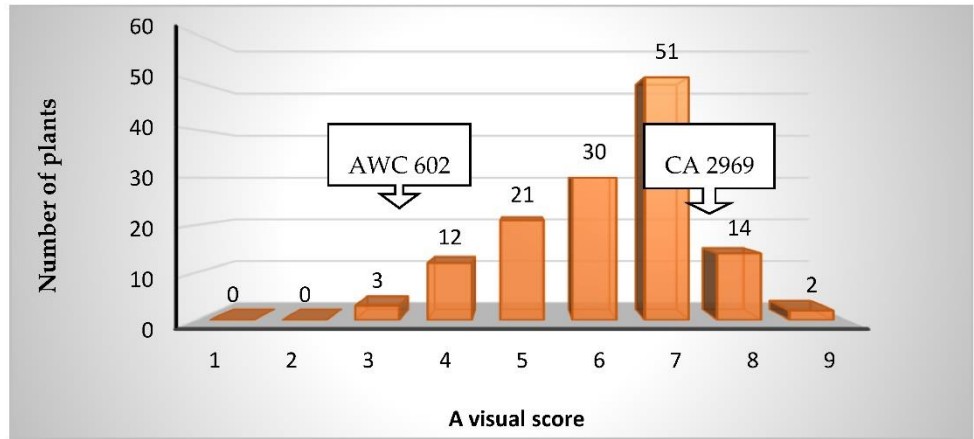

**Figure 2.** Distribution of RILs derived from interspecific crosses between *C. arietinum* and *C. reticulatum*, on the basis of number of plants as resistant or susceptible to the leafminer.

### 3.2. Agro-Morphological Traits

The genotypic effect for agro-morphological traits was statistically significant ($p < 0.05$). Minimum and maximum values for the number of leaves per plant were counted as 31.00 and 176.50, respectively, while the number of leaflets per leaf was found to be 10 and 14. Ranges for leaflet length and leaflet width were detected to be between 0.5–1.3 cm and 0.3–1 cm, respectively. The plant height ranged from 21.50 cm to 48.00 cm. The number of pods per plant was determined to be between 6 and 63. Biological and seed yields per plant were recorded as 1.40–25.68 g and 0.25–11.81 g, respectively. Maximum and minimum values for 100-seed weight were detected as 4.83 g and 29.24 g, respectively (Table 2).

### 3.3. Relationships between Resistance to the Leafminer and Agro-Morphological Traits

A negative and significant correlation was found between RL and plant height (r = −0.314 **) (Table 2), and also plant height had the highest significant negative direct effect ($p = -0.254$ *) on RL via path analysis (Table 3). The highest indirect effect on RL over plant height belonged to the number of leaves per plant ($p = 0.324$ *) and leaflet length ($p = 0.264$ *). After plant height, the highest direct effects on RL were seed yield ($p = -0.223$ *) and growth habit ($p = -0.172$ *) (Table 3). There were significant correlations between RL and leaflet length (r = −0.267 **), and leaflet width (r = −0.234 **) (Table 2). RL score was statistically and significantly negatively correlated with seed yield (r = −0.213 **) (Table 3).

**Table 3.** Path coefficients on direct (bold) and indirect effects of agro-morphological traits on resistance to the leafminer in RILs derived from interspecific crosses between *C. arietinum* and *C. reticulatum*.

| Traits | LP | LL | LH | LW | PH | PP | BY | SY | SW | GH | PA |
|---|---|---|---|---|---|---|---|---|---|---|---|
| Leaves per plant (LP) | **0.012** | 0.028 | 0.060 | −0.014 | 0.324 * | 0.004 | −0.025 | 0.014 | −0.034 | −0.062 | 0.063 |
| Leaflets per leaf (LL) | 0.022 | **−0.004** | −0.006 | 0.029 | 0.050 | 0.023 | 0.071 | −0.156 | 0.044 | −0.044 | 0.032 |
| Leaflet length (LH) | −0.034 | 0.089 | **−0.058** | 0.847 | 0.264 * | 0.075 | −0.199 | 0.020 | 0.363 | −0.154 | −0.620 |
| Leaflet width (LW) | 0.165 | −0.019 | 0.743 | **−0.146** | 0.002 | −0.109 | 0.182 | −0.098 | −0.137 | 0.144 | 0.563 |
| Plant height (PH) | 0.422 | 0.082 | 0.126 | 0.001 | **−0.254 *** | 0.032 | 0.015 | 0.188 * | 0.013 | −0.282 * | −0.128 |
| Pods per plant (PP) | 0.006 | 0.047 | 0.044 | −0.072 | 0.039 | **0.032** | 0.480 | 0.425 * | −0.452 | −0.015 | −0.103 |
| Biological yield (BY) | −0.036 | 0.129 | −0.106 | 0.110 | 0.016 | 0.440 | **0.073** | 0.199 | 0.341 | −0.231 * | −0.189 |
| Seed yield (SY) | 0.021 | −0.286 | 0.011 | −0.060 | 0.212 | 0.392 | 0.200 | **−0.223 *** | 0.349 | 0.083 | 0.209 |
| 100-seed weight (SW) | −0.037 | 0.060 | 0.144 | −0.062 | 0.010 | −0.309 | 0.254 | 0.258* | **0.035** | 0.171 * | 0.164 |
| Growth habit (GH) | −0.056 | −0.051 | −0.051 | 0.054 | −0.196 | −0.009 | −0.144 | 0.052 | 0.143 | **−0.172 *** | −0.167 |
| Pods per axil (PA) | 0.060 | 0.038 | −0.217 | 0.224 | −0.094 | −0.062 | −0.124 | 0.137 | 0.145 | −0.176 * | **0.117** |

\* means that path coefficients are significant at $p < 0.05$ level.

### 3.4. Inheritance Pattern for Resistance to the Leafminer

Heritability for resistance to leafminer in RILs was found to be $h^2 = 30.87$. The prevalence of susceptible lines among RILs indicated that RL was mostly conferred by recessive genes (Figure 2).

### 3.5. Organic Acids

Variation in succinic, malic, citric, tartaric, quinic and oxalic acid levels in leafminer resistant and susceptible RILs in the field (insect infested conditions) and greenhouse (non-infected conditions) were given in Figure S1. Significant differences among RILs were observed for all compounds ($p < 0.05$) in Table S1.

### 3.6. Differences between Organic Acids in Resistant and Susceptible Genotypes of Chickpea

Figure 3 illustrates results for organic acids in resistant and susceptible genotypes of 16 infested chickpea plants under field conditions. Five organic acids including oxalic acid, malic acid, citric acid, quinic acid and tartaric acid did not exhibit any significant differences between susceptible and resistant genotypes under field conditions. However, significant differences were apparent just for succinic acid ($p < 0.05$).

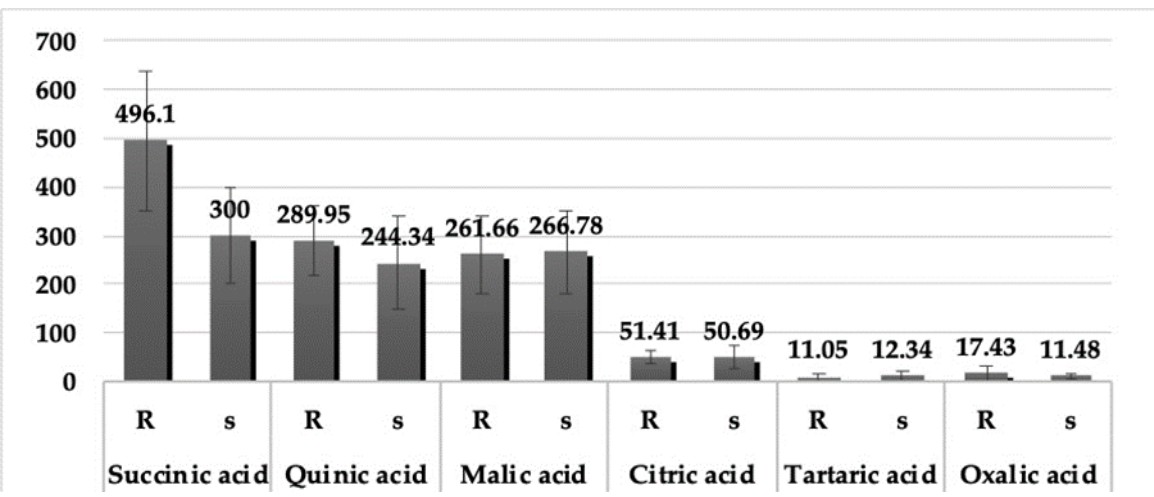

**Figure 3.** Organic acid content in resistant (R) and susceptible (s) RILs of chickpea. Bars indicate means ± standard errors.

Succinic acid amounts were recorded as 496.1 ppm in resistant genotypes, while in susceptible ones these were 300 ppm (Figure 3). Our results showed that the highest organic acid content was succinic acid, followed by quinic, malic, citric and oxalic acids, respectively, while the lowest organic acid content was tartaric acid.

### 3.7. Organic Acids in Chickpea Genotype in Relation to Expression of Resistance to Leaf Miner

Succinic acid was significantly negatively correlated with RL scores (−0.587 **). RL scores also had negative correlations with oxalic acid (−0.264) and quinic acid (−0.260), while malic acid (0.047), citric acid (0.043) and tartaric acid (0.028) were positively correlated with RL scores (Table 4). Path coefficient has been used in determining the character interrelationships and yield criteria for indirect selection [25], and also provides an effective way of finding out direct and indirect sources of correlation. In the present investigation, succinic acid ($p = 0.760$ **) exhibited the highest magnitude of direct effects on RL scores, followed by citric acid ($p = 0.293$) and malic acid ($p = 0.253$). In addition, succinic acid showed significant positive indirect effects via quinic acid ($p = 0.633$ *) and citric acid ($p = 0.341$ *), as shown in Table 4.

**Table 4.** Path coefficients on direct (bold) and indirect effects of organic acids on RL scores in the field and correlations between RL and organic acids.

| Organic Acids | OA | QA | MA | CA | SA | TA | Correlations |
|---|---|---|---|---|---|---|---|
| Oxalic acid (OA) | **−0.304 *** | 0.629 * | 0.081 | 0.124 | −0.208 | −0.265 | −0.264 |
| Quinic acid (QA) | 0.302 | **0.034** | 0.283 | −0.174 | 0.353 * | 0.308 | −0.260 |
| Malic acid (MA) | 0.073 | 0.525 * | **0.253** | −0.104 | −0.069 | 0.043 | 0.047 |
| Citric acid (CA) | 0.135 | −0.393 | −0.127 | **0.293** | 0.429 * | 0.013 | 0.043 |
| Succinic acid (SA) | −0.179 | 0.633 * | −0.067 | 0.341 * | **0.760 **** | 0.030 | −0.587 ** |
| Tartaric acid (TA) | −0.243 | 0.587 * | 0.044 | 0.011 | 0.032 | **0.198** | 0.028 |

\* and ** mean that path and correlation coefficients are significant at $p < 0.05$ and $p < 0.05$ levels, respectively.

Factor analysis was performed and corresponding factor loadings were determined (Table 5). Results revealed that four factors were able to explain 85.4% of the total variation. Factors 1, 2, 3 and 4 explained 35.6%, 21.2%, 15.9% and 12.7% of total variance, respectively. Single evaluation of factor 1 revealed the loadings as 0.899 for quinic acid, 0.649 for succinic acid and 0.646 for malic acid. In factor 2, only RL scores had the highest value with a positive loading (0.659). Factor 3 consisted of oxalic acid with negative loading (−0.658) and tartaric acid with positive loading (0.604), while the last factor consisted of only citric acid with negative loading (−0.699).

**Table 5.** Factor loadings (bold) and communalities of organic acids for principal factors in RILs.

| Source of Variance | Factor 1 | Factor 2 | Factor 3 | Factor 4 | Communality |
|---|---|---|---|---|---|
| Quinic acid | **0.899** | 0.146 | 0.038 | −0.046 | 0.833 |
| Succinic acid | **0.649** | −0.624 | 0.260 | 0.093 | 0.887 |
| Malic acid | **0.646** | 0.482 | 0.002 | −0.211 | 0.694 |
| RL scores | −0.472 | **0.659** | 0.372 | −0.361 | 0.926 |
| Oxalic acid | 0.490 | 0.053 | **−0.658** | −0.458 | 0.887 |
| Tartaric acid | 0.590 | 0.233 | **0.604** | 0.068 | 0.771 |
| Citric acid | −0.187 | −0.590 | 0.330 | **−0.699** | 0.981 |
| Variance | 2.5 | 1.5 | 1.1 | 0.9 | 6.08 |
| Variance% | 35.6 | 21.2 | 15.9 | 12.7 | **85.4** |

## 4. Discussion

Intense infestations of *L. cicerina* were observed in the present study. The susceptible parent CA 2969 and the check Sierra had 7.5 and 9 scores on the visual scale, indicating that climatic conditions and soil properties in the experimental area were appropriate to screen for RL. Cikman et al. [27] have reported that *L. cicerina* produced two generations per year during the spring in the Sanliurfa province of Turkey. The present study suggests that weather conditions in Antalya may also be suitable for the production of two or more generations of leaf miner. A 4 day incubation [2,28,29] time of leafminer under convenient conditions accelerated leafminer infestations. The male parent *C. reticulatum* (AWC 602) and female parent *C. arietinum* (CA 2969) were found to be resistant and susceptible to leafminer with scores of 3.5 and 7.5, respectively, on the visual 1–9 scale, while RIL 39, RIL 56 and RIL 72 were determined to be more resistant to leafminer than the best parent, with a score of 3 on the scale. Twelve RILs were moderately resistant with scores of 3 to 4 on the scale. Superior lines were found not only for RL, but also for agro-morphological traits. Superior lines showing values over those of the parents in both positive and negative directions were observed due to transgressive segregation [30–32]. Superior lines for RL suggested that the resistance level for leafminer could be improved if suitable parents were chosen in interspecific crosses. The number of pods per plant as one of the quantitative agro-morphological traits was found to be higher than that of their parents, demonstrating that the number of pods per plant could be increased via interspecific crosses in chickpea [21,33–35]. Some resistant RILs having a high seed yield (RIL 56 with score of 3.00, RIL 63 and RIL 71 with a score of 4.00), similar to kabuli chickpea,

can be grown directly in the target environments under leafminer infestation conditions. Some resistant and moderately resistant RILs had twin flowers and pods per axil. Using these resistant sources in the field, extra inputs such as insecticides will not need to be used by farmers and products obtained from resistant RILs will be accepted by consumers as good farming practices.

To select for RL, direct and indirect relationships should be considered in chickpea breeding programs. The RILs that were resistant to leaf miner had mostly shorter plant height and smaller leaflet size. Toker et al. [6] stated that there was a close relationship between RL and leaf shape in cultivated chickpea prior to the present study. Sithanantham and Reed [36] pointed out that larger leaflets in chickpea were preferred by leaf miner. The larger leaflet size is considered to be an indicator of more leafminers per leaf [6]. Furthermore, the studies performed with genotypes having multipinnate leaves with narrower leaflets clearly exhibited this situation [11,37]. Almost all the released leafminer-resistant chickpeas and wild chickpeas had multipinnate leaves with small and thin leaflets [8,37,38]. Clement et al. [39] underlined that breeding for polygenic resistance, combining insect repellency (antixenosis), toxicity (antibiosis), and tolerance, could slow the breakdown of plant resistance to chickpea insects. Three categories of insect resistance in legumes were described by Edwards and Singh [40] as follows: (i) structural defenses, (ii) secondary metabolites, and (iii) anti-nutritional compounds. Resistance in the present study could be an effect of antixenosis due to secondary metabolites. A high level of resistance in *C. reticulatum*, as in the present study, was reported for pod borer (*Helicoverpa armigera* Hubner) in some accessions of *C. reticulatum* [41]. The inheritance pattern for RL in RILs was found to be polygenic. Most of the RILs were found to be susceptible to leafminer, showing that RL was governed by polygenes and that these alleles might have provided resistance.

Broad-sense heritability in the RILs was estimated to be $h^2$ = 30.87, explaining that the appearance of resistance might also have been based on additive and genetic effects by year or environment interactions of some alleles because of low level heritability. The result of inheritance was in agreement with the findings for leafminer (*Aproaerema modicella* Deventer) in groundnut [42].

Abiotic stresses such as drought, heat, cold, nutrient deficiency and biotic stresses such as pests, diseases and weeds prevent the realization of the potentially high yield capacity of chickpea [12]. In the Mediterranean area, leafminer causes a reduction in seed yield in chickpea that depends on infestation level, cultivar, and the environment; yield loss rates can reach 40% [2,10,18,37,43]. In addition to physical plant characteristics, the biochemistry of the host plant also exerts profound effects on insect pests by making the host plant less attractive and unsuitable for insect attack.

The composition of organic acids accumulated by plants varies depending upon the species, age of the plant and the tissue type [44]. Some plants use organic acids to cope with nutrient deficiencies, metal tolerance and plant–insect resistance [45].

The present study revealed that succinic acid varied significantly among genotypes and it showed an increase in what might be considered the more resistant genotypes. Michaeli et al. [46] reported that succinic acid plays a role in γ-aminobutyric acid metabolism (GABA), which is a non-proteinogenic amino acid that is found in uni- and multi-cellular organisms and is involved in many responses to stress conditions. Some studies suggest that GABA acts as a player in plant defense by reducing growth and survival of the insects and delaying their development [47,48]. In addition, Bown et al. [49] mentioned that attacked plants had an elevated level of GABA in plant tissue in response to herbivorous insects.

Chickpea has a great capacity to resist the deleterious effects associated with biotic and abiotic stresses due to accumulation of malic acid [50]. In the present study, malic acid showed an interesting pattern: though it was significantly increased under stress, when comparing the levels in the individual genotypes, it showed a decrease in performance of resistant genotypes. These results are contrary to the findings of Bhagwat et al. [51] and

Rembold et al. [52], who reported that varieties with the highest amount of malic acid had the highest resistance to *H. armigera*. Malic acid acted as a deterrent to the *H. armigera* larva. Pod borer-resistant lines had a higher amount of malic acid than the susceptible lines.

Malic acid and succinic acid have been observed to increase in concentration under field conditions compared to greenhouse conditions. This phenomenon can be explained by the fact that a greenhouse trial differs in several aspects to a field trial. It has been noted that the reasons for malic acid and succinic acid accumulation under field conditions may not be due to leafminer stress per se, but rather a result of the stomatal regulatory system response to long-term drought stress [53]. At high photosynthetic rates, malate can accumulate, indicating that the photosynthetic machinery is saturated, thus inducing stomatal closure with the goal of conserving water [54].

In most plants, the main organic acid is malic acid, but tartaric, oxalic or citric acid are predominant [55]. The concentrations of malic acid in different genotypes were relatively high, but tartaric acid was found in low amounts. This result indicated that tartaric acid with low concentration could be a limiting factor for leafminer development. Wang et al. [56] reported a negative effect of tartaric acid concentration on developmental time and offspring survival beginning from egg to adult of *Drosophila suzukii* Matsumura.

Citric acid exhibits an inverse trend in comparison to malic acid (i.e., when citric acid levels decreased under field condition, malic acid levels increased). There is a junction between citric acid and malic acid, the two most abundant acids in plants [57]. Citric acid easily turns into malic acid in the Krebs cycle [58]. It might be possible that the fluctuations in their levels caused a great decrease in citric acid concentrations with the increase in malic acid concentrations [59–61].

Quinic acid was the second highest level in resistant genotypes after succinic acid. Quinic acid is a low molecular weight tannin, and it has been shown to be toxic to several insects; furthermore, it is involved in lignification [62]. Lignin and other phenolics derived from lignin synthesis can strengthen plant cell walls. The strengthened cell walls are difficult to digest and, therefore, can be anti-nutritional for insects [63]. Some research showed that resistant plants accumulated lignin more rapidly and/or exhibited enhanced lignin deposition as compared with susceptible plants [64]. Machado et al. [65] demonstrated that damage caused by herbivore attacks induced more rapid accumulation of quinic acid concentrations in resistant plants than sensitives. These results may indicate a likely participation of quinic acid as a precursor in lignin biosynthesis in chickpea response to leafminer. In accordance, it was found to be accumulated in *Populus* spp. under applied drought conditions [65], so it may be possible that under growth-limiting conditions (such as those that happen during drought), quinic acid accumulated due to lack of utilization in active growth [66,67].

Oxalic acid amounts have been widely studied in chickpea [16,17,37,68]. Oxalic acid and oxalates can provide biochemical and mechanical defense against insect and animal pests [69]. The seemingly simple relationship between biotic stress and the accumulation of oxalic acid may actually be more complex. In fact, oxalates can be toxic to plants, but plants bind those oxalates up in crystals [70]. Furthermore, the leaf oxalic acid content was higher in the least preferred varieties of chickpea leafminer [71]. Oxalic acid was higher in resistant genotypes than susceptible genotypes. This is in contrast to the findings of Cagirgan et al. [15], who reported that oxalic acid in a susceptible genotype of chickpea (ILC 263, susceptible to *Ascochyta rabiei* (Pass.) Labr.) was higher than in resistant genotypes.

For effective selection to improve resistance, it is necessary to have an understanding of various associated traits and the nature of their association with host plant resistance. The association analysis employed in this study provides such required information. The results of correlation analysis between organic acids and RL scores were in accordance with the findings reported by Golla et al. [72], who revealed that leaf surface exudates of chickpea genotypes showed a negative correlation of oxalic acid with oviposition, but a positive correlation with malic acid. Similar results were obtained for oxalic acid in association with leaf damage by *H. armigera* larvae [73]. Patnaik and Senapati [74] reported that egg and

larval counts of pod borer, *H. armigera* were negatively correlated with increasing concentration of acid exudates of chickpea. Low acidity of the chickpea leaf extracts has earlier been reported to be associated with susceptibility to *H. armigera* [75]. Similarly, Selvanarayanan and Narayanasamy [76] reported higher acidity content in resistant tomato genotypes than susceptible check, which expressed significant and negative correlation with both larval population and fruit infestation. Therefore, the simple correlation coefficients are not always effective in determining the real relationships among traits [77]. Path analysis provides an effective way of finding out direct and indirect sources of correlation [78]. The path coefficient analysis offered a slightly similar picture to the simple correlation analysis. The correlation analysis indicated that succinic acid had an important negative influence on RL scores (0.587 **). In addition, succinic acid had the highest magnitude of direct effects on RL scores (0.760 **).

The basic purpose of factor analysis is to generate groups of correlated elements from the initial data set. In the current study, communality values of factor analysis for the measured traits showed that factors one, two, three and four explained 35.6, 21.2, 15.9 and 12.7% of total variance, respectively. The results of factor analysis were most effective when the number of factors was small. Communalities were high (close to 1), and the factors could be readily interpretable in terms of particular sources or processes [79]. Results indicated that correlation analysis between factor and RL scores displayed a strong relationship with factor one in which succinic acid, quinic acid and malic acid had the highest communality and, consequently, the high relative contribution in RL scores.

## 5. Conclusions

Superior lines were found not only for RL, but also for agro-morphological traits, signifying that introgression of resistance genes against leafminer could be possible from *C. reticulatum* to cultivated chickpea via interspecific crosses. The inheritance pattern of resistance to leaf miner in RILs was indicated to be polygenic, and resistance might be based on the addition of some alleles. Three RILs with a score of three (resistant) and 12 RILs with a score of four (moderately resistant) were found to be resistant and moderately resistant to leafminer, respectively. Three of the resistant RILs with a high seed yield resembling the kabuli chickpea can directly be grown in the target environments under leafminer infestation conditions. The use of resistant cultivars could contribute to maintaining sustainable productivity of chickpea without the use of pesticides. The maintenance of such production will ensure both the sustainability of ecological systems and food safety for human health.

After correlation, path and factor analyses, a strong relationship between succinic acid and chickpea leafminer resistance was verified, and such information could be used as a selection criterion for a resistant accession as an integral part of a management program against chickpea leaf miner.

**Supplementary Materials:** The following are available online at https://www.mdpi.com/2073-4395/11/1/57/s1, Figure S1: Citric acid (**A**), malic acid (**B**), quinic acid (**C**), succinic acid (**D**), oxalic acid (**E**) and tartaric acid (**F**) contents (ppm) in resistant (odd-numbered) and susceptible (even-numbered) genotypes under insect-infested (field) and non-infested (greenhouse) conditions. Bars indicate means ± standard errors.

**Author Contributions:** Conceptualization, C.T.; methodology, N.C.; software, H.S.; validation, D.S. and T.E.; formal analysis, M.F.C. and N.C.; investigation, N.C.; resources, C.T.; data curation, H.S., D.S. and T.E.; writing—original draft preparation, N.C., and C.T.; writing—review and editing, M.F.C. and C.I.; visualization, C.T.; consultancy, C.T.; project administration, C.T.; funding acquisition, C.T. All authors have read and agreed to the published version of the manuscript.

**Funding:** This research was funded by the Scientific Research Projects Coordination Unit, Akdeniz University, Antalya, Turkey.

**Acknowledgments:** The authors would like to thank the Scientific Research Projects Coordination Unit for financial support of this project (PhD thesis) under contact no FDK−2018-3743. We also thank the anonymous reviewers for their valuable comments.

**Conflicts of Interest:** The authors declare no conflict of interest.

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
