# Peer review of "Introgression of Resistance to Leafminer (Liriomyza cicerina Rondani) from Cicer reticulatum Ladiz. to C. arietinum L. and Relationships between Potential Biochemical Selection Criteria"

_agronomy, doi:10.3390/agronomy11010057_

Round 1
Reviewer 1 Report
COMMENTS TO REVISED MANUSCRIPT
I could not easily evaluate the Authors' response to my requests, because in their reply they do not refer to all of my points one by one, but only to some of them, and furthermore, they used a new numbering (point 1, Point 2...) that do not correspond to my original line numbering. Also, the line numbers they use in their reply do not match with line numbering I found in the revised manuscript.
The Authors addressed most, but not all of my comments and questions. In some cases they replied but did not change the text accordingly, in other cases they did change the text as required but failed to indicate it in their reply.
In particular, they say: "Response 1: Both parents were homozygous. 119 F2 plant were grown in this interspecific crosses. Single seed descent was advanced that is a seed was harvested each F2 plant. Sierra is a commercial cultivar grown in USA. These information were not given in manuscript since these are not necessary. (Line 97)".
I completely disagree, and kindly ask again to include this information in the manuscript. Readers need this kind of information.
“the recessive or dominant nature of genes conferring RL was predicted on the basis of majority of susceptible and resistant RILs”: this is again not clear. Remove this sentence.
Lines 343-345 has: “Inheritance pattern of resistance to leaf miner in RILs was indicated to be polygenic and governed by minor recessive genes in RILs (Figure 2).”
Lines 1047-1049 has: “Inheritance pattern of resistance to leaf miner in RILs was indicated to be polygenic in nature and governed by some minor recessive genes, and resistance might be based on addition of some alleles”.
Apart from the repetition, this statement is not clear. Maybe the Authors mean that the prevalence of susceptible lines among RILS indicates that leafminer resistance is mostly conferred by recessive genes. This can be included only once, in the results section.
Fig. 3 is not useful to interpret the results. I ask it to move it to supplementary materials, and present in the manuscript a simpler graph reporting the differences between field and greenhouse trials and between the means of the resistant and susceptible lines, and, importantly, the statistical significance of these differences, in a clear way, by integrating Fig. 4.
Fig. 4 shows very large standard errors for succinic acid and my question on statistical significance of the differences between R and s lines was not addressed. The significance of ANOVA and related mean separation test should be clearly reported.
In some cases the Authors misinterpreted my request as in the case of lines 74-75 of revised manuscript, resulting in a wrong sentence: "The control of chicpea leaf miner is most effective when using insecticides weapon to deal with these problems, but they substances have harmful consequences". Also, chickpea is mis-spelled.
I completely disagree with the change in the title (that I did nor request): now it lacks any reference to Cicer: this is not acceptable.
The term "minor genes" should not be used, because it only makes sense as opposed to "major genes". THe size of gene effects are simply not investigated in this work.
The term “seedling stage” is not adequate, it actually indicates “vegetative stage”.
Last, but not least, the quality of the English language is still not adequate, it requires accurate revision for the manuscript to be easily read and understood. Just an example: “the collection of organic acids in the entire plant including leaf, leaflet, stipule and stems is a useful method and easy approach with less time-consuming”.
Fig. 3 A: correct “Non-insected infested”.
Author Response
''Please see the attachment''

Reviewer 2 Report
Thanks for your response.
For response 1, the author can discuss this possibility in the discussion part. Tight linkage between resistance locus and succinic level may not be universal.
For response 2, the author can discuss the future steps and the advantage of MAS after more studies.
For response 3, the author can draw a breeding scheme to show how the RIL was developed. It is easier to follow.
For response 4, really appreciate the detail description of field design with check line. Would the author add this response into the manuscript M&M section please? So readers will know how the field is designed and how infestation was controlled in the experiment.
Author Response
'' Please see the attachment''

Reviewer 3 Report
Kindly find attached my comments
Regards

Author Response
''Please see the attachment''

This manuscript is a resubmission of an earlier submission. The following is a list of the peer review reports and author responses from that submission.
Round 1
Reviewer 1 Report
Review of manuscript submitted to Agronomy:
Introgression of Resistance to Leafminer (Liriomyza cicerina Rondani) from Cicer reticulatum Ladiz. to C. arietinum L. and Relationships between Potential Biochemical Selection Criteria
This manuscript presents data on RILs obtained by crossing a Cicer arietinum line susceptible to Liriomyza cicerina with a C. reticulatum line resistant to the leaf miner. The topic is interesting for chickpea breeders, but in my opinion this manuscript omits much necessary information on materials and methods, is not solid from the statistical point of view, and is not effective in presenting the results. Heritability calculations are not correct. The claim of succinic acid as a possible marker for resistance is weak. Moreover, the discussion extensively repeats results.
My specific comments are the following (numbers are line numbers of pdf file).
27 “investigated potential selection criteria among organic acids”. Change to: “investigated organic acid contents as a selection criteria”
35 Polygenic: change to Quantitative
51 under field conditions: delete
54 but economic damage is unknown: delete
55 “perforate the upside and downside of leaflets to lay their eggs, which number between 1 and 30”. Change to: perforate the leaflet epidermis to lay 1-30 eggs
61 “to deal with these problems but these substances”. Change to: but they
65 “results in ecological imbalance, and destroys not only pests, but also other populations in the ecosystem”. Change to: can result in damage to non-target organisms
67 “that will allow us to continue to fight against insects while reducing the use of chemicals”: delete
75 Were. Change to: lower case.
Species. Change to: lines
76 Improved. Change to: bred
77 “On the other hand, introgression of resistance to the leaf miner from wild C. reticulatum to cultivated chickpea has not been reported in recombinant inbred lines (RILs) derived from interspecific crosses.”
Unnecessary, delete.
82 prey. Change to: pests.
83 “and the exudate contains mainly malic and oxalic acids”: move to previous sentence.
84 “As chemical compounds identified in exudates of chickpea”: unnecessary, delete
99 “A total of 16 RILs 99 were grown in the greenhouse under non-insect conditions and the open field under natural infested 100 insect conditions, and the organic acids were detected by using high performance liquid 101 chromatography (HPLC)”
Unnecessary, delete
MATERIALS AND METHODS
First describe plant materials, then the field trials.
112 non-insect conditions. Change to: non-insect infested conditions
117 “while an attempt was made to control these with irrigation in the greenhouse to prevent drought [24-25]”.
Not clear, temperature is not controlled by irrigation. Delete
120 Level: delete
122 The description of plant materials is not clear and detailed as it should be. Please describe if both parents are homozygous, how the crosses were made, how many F1 plants were obtained, how many F2 plants were used for creating the RILs. Was each RIL derived from a single F2 plant? Is Sierra a commercial cultivar?
128 “in order for there to be a relationship between simple leaf shape and susceptibility to leaf miner in chickpea [6,10,11]”.
Not clear (and not written correctly). Having only one simple-leaf genotype in the trial cannot inform on relationship between leaf shape and resistance.
136 Not clear. Was the trial the same in the field and greenhouse? Almost 80 mq are necessary. Was direct sowing done in the greenhouse as in the field?
160 biological yield: how defined?
166 I do not think that heritability can be calculated with these data. The parents are the same for all RILs, so X and signed-X are constant values for all RILs. The formula of Poehlman and Sleper only works with progenies derived from different pairs of parental plants.
170 “According to distribution for RL, the inheritance pattern was estimated in RILs. Similarly, the recessive or dominant nature of genes conferring RL was predicted on the basis of majority of susceptible and resistant RILs”.
RL is defined at line 197, it should be define here.
Not clear: do you mean that if less than 50% RILs are resistant then a recessive nature of the allele(s) conferring resistance is assumed?
152 “Screening scores between 1 and 4 were resistant, whereas they were categorized as susceptible between 5 and 9”.
Clear cut separation between resistant and susceptible plants for intermediate classes leave doubts.
Wouldn’t be safer, from a breeder’s point of view, taking classes 1-3 as resistant?
174 “during the seedling stage”.
How defined? Were 50 g of leaf tissue sampled per RIL? Were all plant of each RIL sampled?
Ho many replications were performed?
184 “Detector) DAD”, delete
186 Mobil. Change to: Mobile
containing 1 lt. Change to: containing 1 L
191 “Organic acid analyses were used prior to the present study [14].”
If this is the method used, it should be cited before describing them
192 Statistical analyses. More details are necessary on the statistical package and model used for ANOVA, and on softwares used for path and factor analyses.
194 Comprised. Change to: compared
195 Data on resistance to the leafminer. Change to: RL data
197 resistance to the leafminer (RL): this should be defined at line 170
201 resistance to the leafminer (RL). Change to: RL
203 “A total of 130 RILs, their two parents and the check Sierra were identified to be neither very highly resistant nor highly resistant to leaf miner”. Delete.
215 Table 2. The table is not very informative because it only presents the general means. Adding the means of parental genotypes would be useful.
Heritability values are indicated in caption but not presented in the table.
r=-0.141 should not be significant for 129 DF. Please check.
223-229 These results are already in table 2. Delete.
251 “oxalic acid, succinic acid, quinic acid, malic acid, citric acid and tartaric acid”. Change to: all compounds
253-264 This part duplicates information given in fig. 3. Delete.
285 In figure 3, some graphs do not show line 16.
296 What do the bars represent? Standard deviations? They are very long for succinic acid data: are the differences statistically significant?
298-304 Again, this replicates information given in fig.4. Delete
335 It is not clear why four factors were used, and the meaning of the single factors.
346 Table 6. Explain the meaning of bold type.
356 “were determined to be more resistant to leafminer than the best parent with a score of 3 on the scale”.
Significance of this difference is not shown. In fact, the results of the Tukey’s multiple comparison test are not reported.
370-373 “When we searched for RILs in Web of Science (WOS), we encountered a great number of articles [12], yet similarity among RILs for the aforementioned traits such as qualitative traits in these studies had been neither compared nor checked prior to the present study (Figure 3). “
Not pertinent, delete
374-509 The discussion should not repeat results or methods, but only interpret and elaborate on them. This part should be much reduced.
References
The format is not completely homogeneous.
Several references in Turkish language are very difficult to use by readers. If possible, replace with equivalent references in English.
Ref. 2 is 1987, a more recent one would be better.
Author Response
''Please see the attachment''

Reviewer 2 Report
This manuscript described the identification of the correlation between resistance levels to Rondani pest and succinic acid levels in chickpea and suggested succinic level can be used as a biochemical selection marker for Rondani resistance breeding. The data presented in the MS supported these findings. However, several issues need to be addressed.
- This succinic marker may only be used in this specific population. There is coincidence that succinic level is tightly linked to 1 resistance locus in the resistant parent. Results from other population are needed to make a conclusion that succinic could be an universal selection marker in chickpea for Rondani resistance.
- The correlation coefficient between succinic levels and resistance levels is not very high (-.587), as only 16 lines (a very small sample size) were used to make this correlation. It suggested that evaluation of actual resistance is still needed after HPLC evaluation. HPLC's cost is not ideal for practical application in this kind of assistant selection. The authors may consider perform QTL mapping this population and identify DNA markers associated with the resistance. Then the derived DNA marker assisted selection will be more practical than the proposed HPLC' assisted selection.
- The MS failed to present the development of the RIL population.
- The infection was performed under natural infestation conditions. The initial inoculation is not controlled and significant errors may exist.
Author Response
''Please see the attachment''

Reviewer 3 Report
Dear author,
My comments are in the reviewed manuscript attached to this mail. Kindly go through.
Regards

Author Response
Thank you for considering our manuscript for review and giving the opportunity for revision.We appreciate your time to process this manuscript and give review comments immediately . We have carefully taken the comments in consideration to prepare our revision,which has resulted in a paper that is clearer and more compelling.Below are our responses(in blod type) to the comments.The page and numbers refer to our revised manuscript.
Point 1:As plant materials,130 RILs derived from an interspecific cross between cultivated chickpea (C.arietinum,CA 2969) and its progenitor (C.reticulatum AWC602) were screened for resistance to leafminer.
Why 130 RILs or 16 RILs?
Response 1: our study were carried out in natural insect epidemic conditions in 2017 and 2018. In trials,130 RILs,their parents and a genotype susceptible to leafminer (Sierra) as control were used in field conditions.RILs were evaluated for resistance to leafminer using a 1-9 scale when susceptible female parent and Sierra were moderately susceptible (70% insect damage).
According to the field screening results,16 RILs were chosen (eight resistant and 8 susceptible) for growing insect epidemic conditions in the field and non-insect conditions in the greenhouse in order to determine the biochemical (organic acids) selection criteria
Thank you for your valuable comments